# Retrospective Analysis of Clinicopathological Features and Familial Cancer History of Synchronous Bilateral Breast Cancer

**DOI:** 10.3390/healthcare9091203

**Published:** 2021-09-13

**Authors:** Kai-Ling Huang, Yu-Ling Liu, Ya-Ying Hsu, Wen-Ling Kuo

**Affiliations:** 1Department of Medical Education, Chang Gung Memorial Hospital, Linkou and Taipei Medical Center, Chang Gung Medical Foundation, Taoyuan 33305, Taiwan; k.l.huang9@gmail.com; 2Department of Orthopedics, Kaohsiung Veterans General Hospital, Kaohsiung 81362, Taiwan; 3Breast Surgery Division, General Surgery, Department of Surgery, Chang Gung Memorial Hospital, Linkou and Taipei Medical Center, Chang Gung Medical Foundation, Taipei 33305, Taiwan; monayuling@gmail.com; 4Chang Gung University, Taoyuan 33302, Taiwan; softsweethsu@gmail.com

**Keywords:** breast cancer, synchronous, bilateral, germline mutation, cancer gene predisposition, hereditary breast and ovarian cancer syndrome, genetic counseling

## Abstract

Bilateral breast cancer is a strong predictor of BRCA 1/2 mutation and hence one criterion indicated for hereditary genetic testing. The purpose of this study is to assess the characteristics of synchronous bilateral breast cancer (SBBC) and its association with personal and familial cancer traits. Patients diagnosed with SBBC in our institute between 1992 and 2018 were retrospectively reviewed, and the information of clinicopathological features, personal and family cancer history were analyzed. Of the 307 SBBCs enrolled, the growing case number generally aligned with the regional breast cancer incidence after the era of population-based mammography screening. SBBC patients had similar cancer stages but worse survival outcomes than those in the standard scenario. A total of 42.0% had mixed pathological diagnoses, and 22.8% had discordant immunohistochemistry (IHC) subtypes from both sides, which contributed to treatment challenges. The correlation of SBBC with hereditary breast and ovarian cancer (HBOC) syndrome was strongly implied, as 20.7% of our SBBC patients with known familial cancer histories had HOBC-related familial cancers (breast, ovarian, or prostate cancers). These findings highlight the need for genetic counseling and germline mutation testing in patients with SBBC. Early PARP inhibitor treatment should also be considered in high-risk cases for outcome improvement.

## 1. Introduction

Bilateral breast cancer is one of the risk-enrichment criteria for BRCA1/2 mutation in the Pen II risk model [1], BRCAPRO [2], and NCCN guidelines [3] to select appropriate patients indicated for germline mutation testing. Synchronous bilateral breast cancer (SBBC) refers to the simultaneous development of breast cancers in both breasts, while metachronous or asynchronous breast cancer refers to bilateral breast cancer occurring at different times. There is no doubt that SBBC is a strong manifestation of personal predisposition to developing breast cancer.

Women diagnosed with breast cancer have a 2-to-6-fold higher risk of contralateral breast cancer than women at risk of first breast cancer [4]. Bilateral breast cancer accounts for 1–3% of all breast cancers and is a strong predictor for BRCA mutation carriers [3]. Risk factors contributing to bilateral diseases include younger age at diagnosis, presence of family history, germline mutations or single nucleotide polymorphisms, alcohol consumption, and tumor histology, such as lobular carcinoma [5]. In a Korean cohort, 16.3% of patients with bilateral breast cancer had germline BRCA mutations [6]. SBBC patients in a Chinese cohort had higher BRCA 1/2 mutations when there was a presence of family history and bilateral estrogen receptor (ER)-negative disease [7]. In a Polish cohort, BRCA1/2 mutations were present in as many as 29.6% of bilateral breast cancers. Surprisingly, SBBC was less common in BRCA1/2 mutation carriers (12.5%) than in non-carriers (34.2%) [8]. In the post-genomic era, although the ability to confirm and control the genetic impact of personal cancer predisposition has been considerably enhanced, non-genetic causes of bilateral breast cancer or personal breast cancer predisposition remain challenging to determine.

Breast cancer patients with BRCA mutations have a 3% annual risk of contralateral breast cancer, compared to 0.5% in average patients [5]. In a Swedish breast cancer registry-based study from 1970 to 2000, after 10 years of follow up, bilateral breast cancer showed a higher cumulative mortality rate (45%, 95% CI 41.4–48.0%) than unilateral breast cancer (30%, 95% CI, 32.8–33.5%). Metachronous breast cancer developing within 5 years after the diagnosis of the first primary cancer present with a higher mortality rate than metachronous disease developing longer than 5 years apart. Women with bilateral breast cancer younger than 50 years old present with a worse prognosis [9]. Similar results were reported by a study of a Taiwanese breast cancer cohort diagnosed between 1990 and 1999 [10].

SBBC, about 1% by incidence, is associated with a significantly higher distant metastasis rate than unilateral breast cancer and hence worse disease-free (HR 2.6; 95% CI 1.4–4.5) and overall survival (HR 2.3; 95% CI 1.5–2.3) [11]. However, breast cancer incidence rates vary among different ethnic groups and can be affected by cohort transitions. With the implementation of population-wide breast cancer screening, more asymptomatic bilateral cancers can be identified at earlier stages. Treatment according to the molecular subtypes of SBBC involves more complicated strategies but, on the other hand, would significantly change the outcome of bilateral breast cancer. However, from the patients’ point of view, the diagnosis of bilateral breast cancer is still detrimental. Hence, an increasing number of patients with advanced unilateral breast cancer opt to receive bilateral mastectomies after neoadjuvant treatment based on their preference to seek peace of mind [12].

The objective of this study was to review the clinicopathological features of SBBC across a 26-year single institutional cohort, with a focus on the factors related to personal cancer predisposition and the management to mitigate the risks. With the understanding of how SBBC may be related to hereditary breast and ovarian cancer syndrome (HBOC), an effort toward the encouragement of hereditary genetic counseling and testing for SBBC patients can be greatly supported.

## 2. Materials and Methods

With the approval of the Institutional Review Board of Chang Gung Memorial Hospital (IRB No.201800793B0), we retrospectively reviewed all breast cancer cases from the breast cancer registry database in our institute from 1992 to 2018. We defined SBBC as simultaneous diagnosis of bilateral breast cancer and breast cancer from each side diagnosed within 6 months apart, and those meeting the criteria were selected for further analysis. A total of 307 SBBC patients were enrolled in our study. Clinical pathological parameters, including age at diagnosis, menstruation history, pathological diagnosis, anatomical staging, surgical treatment, and follow-up status were collected until December 2019. Breast cancer subtypes were defined based on the receptor status by tumor immunohistochemical staining (IHC), including hormone receptor (HR) and human epidermal growth factor receptor type 2 (HER2) with or without fluorescence in situ hybridization (FISH) for the detection of amplified HER2 gene. Familial and personal cancer history was particularly extrapolated from electronic medical records to compare the outcomes of patients with or without a family history.

Statistical analyses were performed using IBM SPSS Statistics for Windows, version 20.0. (Armonk, NY, USA: IBM Corp). Chi-square or Fisher’s exact test was used to compare the relevant parameters. All tests in our study assumed a 95% confidence interval (CI), and *p* < 0.05 was considered statistically significant. Survival was compared with Kaplan–Meier analysis with log-rank test. 

## 3. Results

Between 1992 and 2018, a total of 16,982 breast cancers were treated in our hospital. Among them, 307 patients (614 affected breasts, 3.6% of all breast cancers treated in the same period) had simultaneous diagnoses of bilateral breast cancers, or occurrence of breast cancer on each side of the breast within 6 months in the same individual according to the definition in our study. Their clinicopathological characteristics are shown in Table 1. The median age was 52 years (range 31–83), with 56.4% of patients aged 50 and above, and 46.9% were postmenopausal at the time of diagnosis; at the time of diagnosis, 60.0% of patients were parous, 14.3% were nulliparous, and 25.7% were unknown. The trend of bilateral breast cancer diagnosis correlates with the incidence of breast cancer both in our hospital and the national registry (Figure 1) [13], with the number of SBBC increasing steadily from 1–3 cases per year (0–1.8%) before 2002 to 41 cases per year (7.5%) in 2017. There was a similar distribution of symptoms in the right (22.1%), left (17.4%), and bilateral breasts (14.3%), but some experienced no symptoms at all (28.6%). An increase in the number of SBBC cases became significant after 2004, when the breast cancer public screening program was initiated. 

The information about the presence of symptoms was unavailable in 35% of our patients from their medical records, but at least 28.6% (88 cases) of our SBBC patients had clear documentation of being asymptomatic. This corresponds to the same proportion of patients diagnosed at stage 0 (6.2%) and 1 (26.4%). Many (59.6%) patients showed low to intermediate tumor grade (grade 1 or 2) in bilateral breasts, and about half (48.9%) had no axillary lymph node involvement; the remainder had unilateral (*n* = 113, 36.8%) and bilateral (*n* = 44, 14.3%) axillary lymph node involvement. About two-thirds of our cases (65.2%) had less advanced breast cancer, with stage 0, stage I, and stage II accounting for 6.2%, 26.4%, and 32.6%, respectively. Of note, 14 diseased breasts (2.3% of all diseased breasts, accounting for 9 patients (3.0% of all patients)) had inflammatory breast cancer. The same pathological diagnosis for both breasts was found in 57.7% of patients, with 46.2% IDC, 7.5% DICS, 3.6% ILC, and 0.3% mucinous cancer. The remaining patients had a different pathological diagnosis from each breast, with DCIS-IDC occurring most frequently (30.9%), followed by IDC-ILC (4.2%) and IDC-other (2.9%) (Table 1). Regardless of the stage of diagnosis, mastectomy (363 affected breasts, 184 on the left and 179 on the right, 59.1% of all diseased breasts) was preferred over breast conservation surgery (BCS) (185 affected breasts, 90 in the left and 95 on the right, 30.1% of all affected breasts), with 51.8% of patients receiving bilateral mastectomies, and 85.3% (51.8% bilateral mastectomy, 23.1% bilateral BCS, 10.4% bilateral non-surgical treatment) of patients had the same surgical management for both breasts. Another 10.4% of patients received non-surgical treatment on both sides (Table 2).

IHC results were available for 88% of cancer cases in our SBBC cohort. In patients with available IHC results, the HR+ HER2− subtype was the most common (70.3%), followed by HR+ HER2+ (12.2%), HR− HER2− (7.8%), and HER- HER2+ (9.6%). A total of 56.7% (174 cases) of SBBC patients had concordant IHC expression in bilateral breasts, while 22.8% had different IHC profiles from both sides. In patients with concordant expression in bilateral breast cancers, the majority were HR+ HER2− (143 patients, 82.2% out of 174 patients with concordant expression), followed by HR+ HER2+ (13 patients, 7.5%), HR− HER2− (11 patients, 6.3%) and HR− HER2+ (7 patients, 4.0%) (Table 3). In patients with disconcordant expression, the top three combinations by order of occurrence were HR+ HER2−/HR+ HER2+ (34.3%), HR+ HER2−/HR− HER2− (28.6%), HR+ HER2−/HR− HER2+ (21.4%) (Table 3). The remaining 20.5% of patients had incomplete IHC results for bilateral breasts; thus, their concordance remains unknown.

Only 4.2% (*n* = 13) of all SBBC cases had record of other primary cancers in addition to breast cancer (Table 4), namely lung cancer (*n* = 3), ovarian cancer (*n* = 2), thyroid cancer (*n* = 2), endometrial cancer (*n* = 2), colon cancer (*n* = 2), hepatocellular carcinoma (*n* = 1), and melanoma (*n* = 1) (Figure 2a). Familial cancer history was documented in 22.8% of all SBBC patients, the top three being breast cancer (*n* = 54, 17.6% of all SBBC), hepatocellular carcinoma (7.1%), and prostate cancer (5.7%) (Figure 2b). By the exclusion of 22 cases with unknown familial cancer history, 20.7% (59 out of 285 SBBC patients with available family history information) had a family history of breast, ovary, or prostate cancer, suggestive of HBOC. HBOC-related cancers represented 84.3% (59 cases out of 70) of cases with positive familial cancer history.

After a mean follow-up of 59.2 months, the 5-year disease-free survival rates of patients with stage 0 to 3 SBBC were 85.3%, 88.4%, 85.6%, and 71.8%, respectively, and the overall survival rates for all patients were 100%, 97.7%, 97.1%, 81.2% and 55.5% for stages 0,1,2,3,4, respectively, showing a statistically significant difference (*p* < 0.001) (Table 5). These outcomes of SBBC were inferior to those common for unilateral breast cancer.

## 4. Discussion

Across a study period of 26 years, 307 patients with SBBC were enrolled and analyzed in our cohort. To our knowledge, this is one of the largest case series focusing on SBBC from a single institution. The age of onset, pathological diagnosis, and stage at diagnosis were similar to those for unilateral breast cancer. The proportion of breasts affected with triple-negative breast cancer was 9.6%, comparable to the general population of unilateral breast cancer. 

The majority of SBBC cases had early disease at diagnosis in our series, with 48.9% without axillary lymph node metastasis, 59.6% with low to intermediate grade, and 65.2% diagnosed before stage 2. A noteworthy finding is that only 14.3% of patients had symptoms in bilateral breasts, while at least 28.6% presented with no symptoms in both breasts, highlighting the necessity of population-based mammography screening, which helps detect asymptomatic or occult bilateral breast cancer that could have been missed. Although the lobular phenotype, including lobular neoplasm and invasive lobular carcinoma (ILC), is generally considered to be associated with bilateral breast cancer [12,14], only 36 (11.7%) cases in our cohort had a pathological diagnosis of lobular histology.

Patients with increased breast cancer predisposition often present with early disease onset, bilateral breast cancer, or multiple cancers in a single individual or within the family. A high risk for breast cancer is viewed as an increased predisposition to develop breast cancer, as explained by genetic or non-genetic factors. Genetic factors include hereditary mutations in high penetrance genes, such as BRCA1 or 2, resulting in hereditary breast and ovarian cancer (HBOC), while other predisposition genes such as PALB2, CDH, PTEN, ATM, STK11, CHEK2, and TP53 have lower penetrance [15]. Non-genetic factors include lifestyles with increased exposure to environmental carcinogens and metabolic adaptation [16], resulting in upregulated signaling pathways or microenvironments supporting tumor growth. Epigenetic regulation causing oncogene activation or malfunction of tumor suppressor genes [17] may also play important roles. 

Although we do not have genomic results from these patients, the impact of genetic disposition cannot be easily overlooked in patients with SBBC. As many as 20.7% of patients with SBBC presented with a family history of breast, ovarian, or prostate cancer in our cohort, suggestive of HBOC, or germline BRCA mutation could be the primary cause of their cancer predisposition. According to the NCCN guidelines for hereditary cancer testing [3], they would meet the criteria to test for high-penetrance breast or ovarian cancer susceptibility genes. The prevalence of mutations in these genes in the overall breast cancer population is approximately 5%, and the threshold for testing is an estimated 10% carrier rate in candidate populations. For those without HBOC syndrome, other cancer gene predispositions may also be discovered with multigene panels. 

Ways to manage the risk of cancer development include enhanced surveillance, regular screening, and preventive measures to avoid cancer occurrence. Taking HBOC as an example, risk-reducing salpingo-oophorectomy (RRSO) reduces all-cause mortality in germline BRCA 1/2 mutation carriers with (HR, 0.43; 95% CI 0.318–0.588) or without breast cancer (HR, 0.34; 95% CI, 0.190–0.639) [18,19], and has been the only risk-reducing measure recommended by the NCCN (National Comprehensive Cancer Network) guidelines [3]. Bilateral risk-reducing prophylactic mastectomy (BRRM) significantly reduces breast cancer incidence, but is only regarded as a personal option because it is not associated with improved all-cause mortality [18]. Moreover, the side effects such as disfiguration, sensory loss of the skin or nipple, and the cost of reconstructive surgery limit the benefit of BRRM only to well-defined high-risk patients bearing a strong genetic predisposition [18].

The most well-known high-penetrance breast cancer predisposition genes are BRCA1 and BRCA2, each predicting 84% and 56% lifetime breast cancer risk [20]. Although the prevalence of germline BRCA mutation carriers in Asian countries is believed to be lower than in Western countries or among Ashkenazi Jewish descents, the actual percentage remained elusive before the post-genomic era. Recent reports have identified as high as 11.1% germline BRCA1 mutation in a risk-enriched cohort of Taiwanese breast cancer patients [21]. In another Korean study, the prevalence of BRCA mutations in specific breast cancer subgroups was 4.8% in male patients, 8.8% in early- onset (<35 years) patients without a family history, 16.3% in patients with bilateral breast cancers, 22.4% in patients with a family history of breast or ovarian cancer, and 37.5% in patients with both breast and ovarian cancer [6].

Germline mutation testing has more clinical value than that before 2021. The poly(adenosine diphosphate-ribose) polymerase (PARP) inhibitor, olaparib, and talazoparib have shown promising results in the neoadjuvant [22], adjuvant [23], and advanced breast cancer setting in patients with a germline BRCA mutation [24,25]. Thus, the most updated NCCN guidelines [26] have suggested its early use in the adjuvant treatment of high-risk patients with pathogenic or likely pathogenic germline BRCA mutations to reduce the invasive disease recurrence rate by 8.8% at 3 years. The results of our study further justify genetic testing in patients with early SBBC, as a substantial percentage of patients may benefit from the use of such novel agents. 

Our study has several limitations. It is a retrospective study, prone to selection bias and missing data, and alterations in treatment standards, especially over a period spanning 26 years. Obtaining family history from electronic records was a major challenge, and we highly speculate an underestimation of family history, especially HBOC-related, in these SBBC patients. Further investigation to include patients with metachronous bilateral breast cancer may further establish the characteristics of bilateral breast cancer and its relationship to genetic disposition. 

## 5. Conclusions

Our study shows that many patients with SBBC are asymptomatic or screen-detected, and often have different IHC expression or pathological diagnoses in bilateral breasts. Meticulous evaluation of the contralateral side in patients with unilateral breast cancer is helpful. Up to 20% of patients with SBBC may have HBOC-related family history, suggesting a strong correlation to cancer gene predisposition, and should be offered genetic counseling and germline mutation testing.

## Figures and Tables

**Figure 1 healthcare-09-01203-f001:**
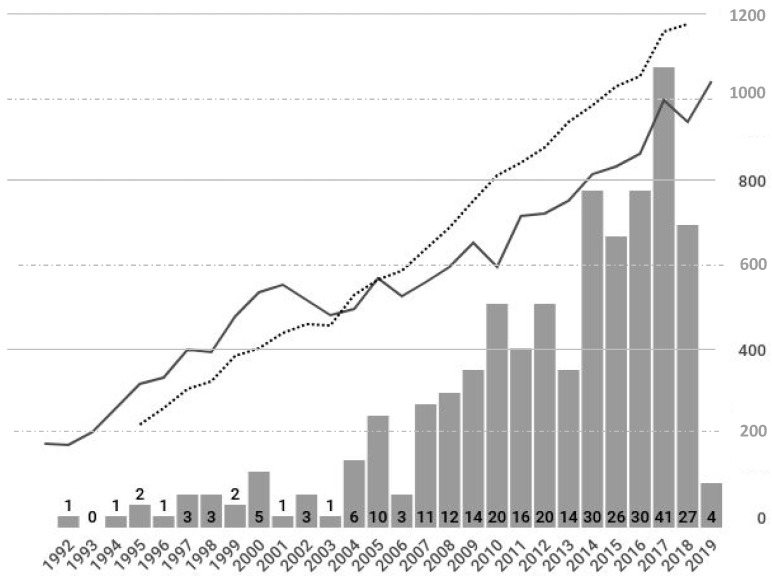
The incidence trends of synchronous bilateral breast cancer per year (bar with numbers), number of breast cancer at our institution per year (solid line, right side y-coordinate), and national incidence rate per 10,000 population (dotted line, also right side y-coordinate) in Taiwan from the Health Promotion Administration, Ministry of Health and Welfare (HPA) breast cancer registry.

**Figure 2 healthcare-09-01203-f002:**
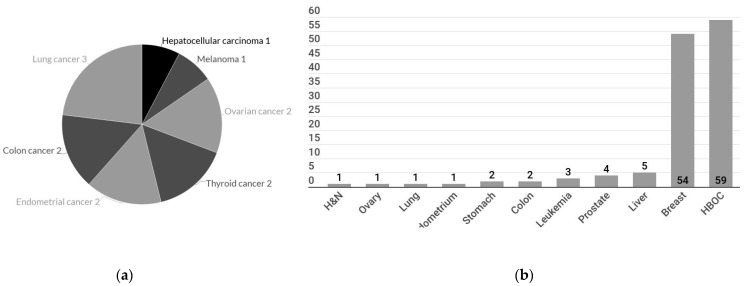
Personal and family cancer history. (**a**) Personal history other than breast cancer. (**b**) Familial cancer history in patients with synchronous bilateral breast cancers. H&N: Head and neck cancer. HBOC (hereditary breast and ovarian cancer syndrome): includes patients with familial cancer history of breast, ovary, or prostate.

**Table 1 healthcare-09-01203-t001:** Clinicopathological characteristics of synchronous bilateral breast cancers.

Characteristic		Median (Range) or *n* (%)
Age at time of diagnosis	Median, years	52 (31–83)
	<50 years old	134 (43.6%)
	≥50 years old	173 (56.3%)
Age of menarche	Median, years	14 (11–19)
Menopausal Status	Premenopausal	103 (33.6%)
	Postmenopausal	144 (46.9%)
	Perimenopausal/unknown	60 (19.5%)
Childbearing	Parous	184 (60.0%)
	Nulliparous	44 (14.3%)
	Unknown	79 (25.7%)
Initial symptomatic breast	Right	68 (22.1%)
	Left	54 (17.4%)
	Bilateral	44 (14.3%)
	Asymptomatic	88 (28.6%)
	Unknown	53 (17.4%)
Pathological diagnosis	Bilaterally concordant	177 (57.7%)
	IDC	42 (46.2%)
	ILC	11 (3.6%)
	DCIS	23 (7.5%)
	Mucinous	1 (0.3%)
	Bilaterally disconcordant	129 (42.0%)
	IDC-DCIS	95 (30.9%)
	ILC-DCIS	6 (2.0%)
	IDC-ILC	13 (4.2%)
	DCIS-LCIS	2 (0.7%)
	IDC-LCIS	1 (0.3%)
	IDC-other ^1^	9 (2.9%)
	ILC-other ^2^	3 (1.0%)
	Unknown	1 (0.3%)
Tumor grade	Bilateral grade 3	29 (9.4%)
	Unilateral grade 3	56 (18.2%)
	Bilateral grade 1/2	183 (59.6%)
	Unknown/Other	39 (12.7%)
Tumor stage ^3^	Stage 0	19 (6.2%)
	Stage 1	81 (26.4%)
	Stage 2	100 (32.6%)
	Stage 3	69 (22.5%)
	Stage 4	37 (12.1%)
	Unknown	1 (0.3%)
Axillary lymph node metastasis	Bilateral	44 (14.3%)
	Unilateral	113 (36.8%)
	None	150 (48.9%)

^1^ Other pathological diagnoses include (in order of frequency): mucinous carcinoma (*n* = 4, 1.3%); metaplastic carcinoma (*n* = 2, 0.6%), tubular carcinoma (*n* = 1, 0.3%), papillary carcinoma (*n* = 1, 0.3%) and apocrine carcinoma (*n* = 1, 0.3%). ^2^ Other pathological diagnoses include (in order of frequency): metaplastic carcinoma (*n* = 2, 0.6%); mucinous carcinoma (*n* = 1, 0.3%). ^3^ The more advanced stage from both sides. IDC: invasive ductal carcinoma; ILC: invasive lobular carcinoma; DCIS: ductal carcinoma in situ; LCIS: lobular carcinoma in situ or lobular neoplasm.

**Table 2 healthcare-09-01203-t002:** Surgical methods performed for each side in synchronous bilateral breast cancer patients.

Right Breast					
		Mastectomy	BCS	Non-Surgical	Total
Left Breast	Mastectomy	159 (51.8%)	19 (6.1%)	1 (0.3%)	179 (58.3%)
	BCS	24 (7.8%)	71 (23.1%)	0 (0%)	95 (31.0%)
	Non-Surgical	1 (0.3%)	0 (0%)	32 (10.4%)	33 (10.7%)
	Total	184 (60.0%)	90 (29.3%)	33 (11.0%)	307 (100%)

**Table 3 healthcare-09-01203-t003:** Immunohistochemical subtypes of bilateral breast cancers.

Characteristic	Description	*n* (%)
IHC by the number of the affected breasts	Unknown	75 (12.2%)
	IHC available	539 (87.8%)
	HR+ HER2−	379 (70.3%^) 1^
	HR+ HER2+	66 (12.2%) ^1^
	HR− HER2+	42 (7.8%) ^1^
	HR− HER2−	52 (9.6%) ^1^
	Total	614 (100%)
	Bilaterally concordant expression	
IHC by the number of the affected patients	HR+ HER2−	143 (82.2%) ^2^
	HR+ HER2+	13 (7.5%) ^2^
	HR− HER2+	7 (4.0%) ^2^
	HR− HER2−	11 (6.3%) ^2^
	Total concordant cases	174 (56.7%)
	Bilaterally disconcordant expression	
	HR+ HER2−/HR+ HER2+	24 (34.3%) ^3^
	HR+ HER2−/HR− HER2+	15 (21.4%) ^3^
	HR+ HER2−/HR− HER2−	20 (28.6%) ^3^
	HR+ HER2+/HR− HER2+	3 (4.3%) ^3^
	HR+ HER2+/HR− HER2−	5 (7.1%) ^3^
	HR− HER2+/HR− HER2−	3 (4.3%) ^3^
	Total discordant cases	70 (22.8%)
	Unknown^4^	63 (20.5%)

^1^ The percentage out of all IHC available cases. ^2^ The percentage out of bilaterally concordant cases. ^3^ The percentage out of bilaterally known cases. ^4^Patient with unavailable IHC results from either side of the diseased breast is considered unknown, even if the IHC result on one side is known.

**Table 4 healthcare-09-01203-t004:** Personal and family cancer history (*n* = 307, total number of SBBC patients).

Cancer History		*n* (%)
Personal	Known	302 (98.3%)
	With cancer history	13 (4.2%)
	Without cancer history	289 (94.1%)
	Unknown	5 (1.6%)
Familial	Known	285 (92.8%)
	With cancer history	70 (22.8%)
	Breast, ovary, or prostate cancer	59 (19.2%)
	Other cancer types	11 (3.5%)
	Without cancer history	215 (70.0%)
	Unknown	22 (7.2%)

**Table 5 healthcare-09-01203-t005:** Disease-free survival and overall survival of synchronous bilateral breast cancer.

Survival	Stage ^1^	Event (%)	Mean Survival (Months, 95% CI)	5-Year Survival (%)	*p*-Value
Disease-free survival	Stage 0	3/19 (15.7%)	150.809 (109.560–192.057)	85.3%	0.124
	Stage 1	8/81 (9.9%)	208.520 (168.813–248.228)	88.4%	
	Stage 2	16/100 (16%)	221.792 (190.726–252.858)	85.6%	
	Stage 3	17/51 (33.3%)	175.703 (148.917–202.489)	71.8%	
Overall Survival	Stage 0	1/19 (5.2%)	235.539 (235.529–235.529)	100%	<0.001
	Stage 1	3/81 (3.7%)	232.416 (194.108–270.724)	97.7%	
	Stage 2	5/100 (5%)	263.129 (242.890–283.368)	97.1%	
	Stage 3	10/69 (14.5%)	199.920 (176.751–223.643)	81.2%	
	Stage 4	10/38 (26.3%)	121.101 (86.273–155.929)	55.5%	

^1^ Stage refers to the more advanced stage from both sides.

## Data Availability

There are no publicly archived datasets related to this study.

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
