# Peer review of "Retrospective Analysis of Clinicopathological Features and Familial Cancer History of Synchronous Bilateral Breast Cancer"

_healthcare, 2021, doi:10.3390/healthcare9091203_

Round 1
Reviewer 1 Report
This is an interesting manuscript although it would be more interesting if genetic studies were available. I would expect more young patients and advance disease if all SBBC would be related with germline mutations.
Comments:
1) It would be interesting if the authors can add the number of parous and nulliparous patients.
2) Do the authors know how many patients had inflammatory breast cancer (IBC) if any?
3) In Results, line 2, instead of "(614 affected breast)", add the percentage of SBBC from the total cases studied that will be 3.3%.
Reviewer 2 Report
This paper would greatly benefit from the addition of an author who is a statistician.
Figure 1 is a good graph but would benefit from the right sided y-axis being in whole numbers. e.g. Increments of 200...?
Table 1
- Numbers are not pulled out appropriately from this into the text. e.g. "Approximately 28.6% of SBBC patients were asymptomatic" - No, because 17.4% were unknown. Actually, 35% of the known were asymptomatic. This would be a more appropriate number to state and more useful to the reader.
- When quoting 74.9% are stage 0 to 2, I cannot see how this number is derived for the table. In fact, none of the numbers in this paragraph are easily derivable from the table.
Table 2
- Again, the numbers in the text do not tally with the numbers in the table. e.g. 59.1% had mastectomy. Where does this come from? And what denominator are they suggesting it is out of? From all 307? Or just those who had surgery? Denominators must be made clear in each result.
- There are also errors in the table itself.
- e.g. (1) 90/307 having a BCS in the right breast = 29%, not 6.1%
- e.g. (2) 33 patients had Non-surgical in the right breast, not 34.
Table 3
- The first sentence in the paragraph relating this table makes it sound like the following %s will be out of those known. They are not, and thus it is misleading.
- Again, cannot derive the numbers in the text from the table. e.g. 82.1%...?
Table 4
- Is the group "History of breast, ovary or prostate cancer" a sub-group of Yes? If so, this needs to be presented as such. At the moment, it looks like a separate level.
Figure 2a
- The text states all the legend apart from lung...?
- It would be better to order these by prevalence.
Figure 2b
- It would be better to order these by prevalence.
Table 5
- It is fine to state there is a median follow-up of 59.2 months but, as you state the screening programmes have changed over the years, you would thus expect longer follow up on the older (more high grade) cancers and thus FU will be different, and thus needs presenting, split by stage.
- You cannot quote DFS rates "after a median FU of 59.2 years" unless the FU rates are similar over the stages. They are quoting 5-year DFS rates. This must be clearly stated. These %s should have been obtained from Kaplan Meier curves but it is unclear if they were...?
- Where are the stage 4 details?
- It is not clear if the statistical tests applied are appropriate for the data. Log rank tests should be used on the median survival data, not chi squareds on the simple 5-year rates. I am not sure which have been used.
